# A Meta-Narrative Review of Channelopathies and Cannabis: Mechanistic, Epidemiologic, and Forensic Insights into Arrhythmia and Sudden Cardiac Death

**DOI:** 10.3390/ijms26178635

**Published:** 2025-09-04

**Authors:** Ivan Šoša

**Affiliations:** Department of Anatomy, Faculty of Medicine, University of Rijeka, 51000 Rijeka, Croatia; ivan.sosa@uniri.hr

**Keywords:** arrhythmia, cardiac ion channels, cannabinoids, CB1/CB2 receptors, channelopathies, forensic toxicology, sudden cardiac death, Synthetic Cannabinoid Receptor Agonists

## Abstract

Although cannabinoids have proven therapeutic benefits, they are increasingly known for their capacity to disturb cardiac electrophysiology, particularly in individuals with hidden genetic issues such as channelopathies. This review consolidates molecular, clinical, epidemiological, and forensic findings linking cannabinoid exposure to arrhythmias and sudden cardiac death. It examines how phytocannabinoids, synthetic analogs, and endocannabinoids influence calcium and potassium currents through cannabinoid receptor-dependent and -independent pathways, affect autonomic regulation, and contribute to adverse conditions such as oxidative stress and inflammation in heart tissue. Genetic variants in key genes linked to SCD (*SCN5A*, *KCNH2*, *KCNQ1*, *RYR2*, and *NOS1AP*) can reduce repolarization reserve, transforming otherwise subclinical mutations into lethal substrates when combined with cannabinoid-induced electrical disruptions. Forensic research highlights the importance of comprehensive toxicological testing and postmortem genetic analysis in distinguishing between actual causes and incidental findings. There is an urgent need to re-evaluate the cardiovascular safety of cannabinoids, and this is underscored by the findings presented. The merging of molecular, clinical, and forensic evidence reveals that cannabinoid exposure—especially from high-potency synthetic analogs—can reveal latent channelopathies and precipitate fatal arrhythmias. Accordingly, this review advocates for a paradigm shift toward personalized risk stratification. If genetic screening is integrated with ECG surveillance and controlled cannabinoid dosing, risk assessment can be personalized. Ultimately, forensic and epidemiological data highlight the heart’s vulnerability, emphasizing its role as a target of cannabinoid toxicity and as a crucial aspect of public health monitoring.

## 1. Introduction

Cannabinoids are substances particularly well known as herbal compounds that are used recreationally; they interact with the endocannabinoid system and have gained considerable attention for their medicinal benefits [1,2,3]. As regulations develop and cannabinoid use increases, it is essential to consider the broader health implications [4], for example, the potential association between cannabinoids and arrhythmias and sudden cardiac death (SCD) [5,6]. Genetic factors can affect how a human heart’s electrical system operates, potentially increasing a person’s risk of adverse reactions to cannabinoids. Forensic and legal medicine has seen these exact features rise in prominence [7].

Cannabinoids have a significant impact on the electrophysiological properties of cells, primarily through their interaction with cannabinoid receptors and through various ion channels [8,9]. Interactions influence cardiac rhythm by impacting neuronal excitability, specifically synaptic transmission [10]. This makes cannabinoids both therapeutically promising and potentially hazardous [1,2].

In a positive respect, some cannabinoids modulate calcium and potassium channels and have demonstrated neuroprotective and anticonvulsant properties [11,12]. These properties facilitate cannabinoid involvement in regulating (modulating) neuronal firing, and such modulation can reduce hyperexcitability, offering relief in conditions like epilepsy [13]. Cannabinoids also affect sodium channels, thus contributing to their analgesic effects by reducing nociceptive signaling [8].

Nevertheless, the electrophysiological side effects of cannabinoids are considerable and require careful attention. For example, Δ9-tetrahydrocannabinol (THC) has been associated with alterations in cardiac ion-channel function and prolonged QT intervals in some reports, raising concerns about the arrhythmia risk, including QT prolongation and ventricular tachycardia [6]. Chronic cannabinoid exposure may desensitize CB1 receptors, leading to altered neurotransmitter balance and impaired synaptic plasticity [10]. In individuals with underlying genetic channelopathies—such as mutations in *SCN5A* and *KCNQ1*—cannabinoid exposure has been linked to symptom worsening and, in isolated cases, to seizures or sudden cardiac events [14,15]. As voltage-gated sodium channels interact with CBDs, this interaction could provide therapeutic advantages in treating myotonia [16], albeit carrying the risk of changing muscle excitability [8].

The lipophilic nature of these substances and their postmortem redistribution additionally complicates the postmortem process [17]. For these reasons, in addition to the electrophysiological effects, attention in forensic and legal contexts has increasingly been drawn to cannabinoids, which are now recognized as potential contributors in sudden-death investigation [7,18]. In forensic toxicology, the detection of cannabinoids—especially synthetic variants—poses challenges due to their structural diversity and ability to evade standard drug screens [19]. Advanced techniques like liquid chromatography–mass spectrometry (LC-MS/MS) are thus often required to identify these compounds [20].

This review combines existing knowledge on how cannabinoid exposure can lead to adverse heart outcomes through genetic differences, describes key mechanisms, and outlines future research directions.

### 1.1. Overview of Cannabinoids, Biphasic Effect, and Cardiac Ion Channel Modulation

Cannabinoids are a broad category of compounds, including phytocannabinoids from cannabis, synthetic versions, and naturally occurring ligands [12,21]. They primarily act through cannabinoid receptors (CB1 and CB2), which are abundant in tissues like the central nervous system (CNS) and the heart [9]. When activated, these receptors can influence autonomic regulation, vascular tone, and cardiac muscle contractility, and act in a so-called “biphasic manner”. In this phenomenon, low and high doses of the same substance can produce opposite or contrasting effects [22]. At lower doses, cannabinoids may have sedative and anti-inflammatory benefits, but at higher doses or in sensitive individuals, they can cause tachycardia, hypertension, and autonomic instability (Table 1; Figure 1). This is especially risky for people who are prone to arrhythmia, as autonomic fluctuations might trigger dangerous electrical disturbances [23].

Certain cannabinoids appear to provide cardioprotective effects at low doses or under specific conditions [10,11]. Some low-dose cannabinoids—especially CB_2_-selective agonists (e.g., JWH-133), cannabidiol (CBD), and endocannabinoids (anandamide, 2-AG)—offer cardioprotection by dampening inflammation, oxidative stress, and apoptosis through CB2 receptor activation and multi-target signaling [8,11,28]. Administered around ischemia–reperfusion, they reduce infarct size and arrhythmias [10,12]. CBD further enhances endothelial function and limits post-infarct remodeling via its antioxidant, vasodilatory, and anti-fibrotic actions [13,28].

The activation of receptors by cannabinoids modulates various signal transduction pathways, including those that affect ion channel function. Experimental studies suggest that cannabinoids may influence calcium and potassium current dynamics through receptor-mediated or direct non-receptor mechanisms [8]. It has been demonstrated that cannabinoids can alter ion channel function directly or indirectly through receptor-mediated pathways. For example, the activation of CB1 receptors has been linked to the modulation of calcium currents in the context of myocardial contraction and electrophysiological stability. Conversely, this effect is particularly emphasized for its crucial role in the retina [29]. However, at higher concentrations or with prolonged exposure, they may activate harmful pathways, including oxidative stress or inflammation [21,28].

Modifications and interactions like these, though often subtle, can lead to changes in electrophysiological properties such as action potential duration and refractoriness. These factors are essential for maintaining a normal heart rhythm. In other words, they can reduce the threshold for arrhythmia when paired with a proarrhythmic environment resulting from genetically caused impaired ion channel functioning. Additionally, the biphasic effects of cannabinoids on the autonomic nervous system (ANS) may further disrupt heart rhythm by altering the balance between sympathetic and parasympathetic activity. Beyond genetics, these dual effects contribute to the unpredictability of cardiac health.

Because individuals process cannabinoids differently [30], tighter dosage controls or personalized approaches based on genetic and physiological factors may be necessary. In healthy individuals, these effects may be mild or protective in some instances. However, for those with genetic channelopathies—where ion channels already do not work correctly—even minor cannabinoid-induced changes can disrupt electrophysiological stability, risking arrhythmias [6]. This interaction can cause cannabinoids to act as triggers, revealing hidden channel problems that would otherwise stay silent [31]. Building upon these foundational insights, recent studies have elucidated the molecular mechanisms by which cannabinoids modulate cardiac ion channels via CB1 and CB2 receptors. When activated by cannabinoids, CB1 and CB2 receptors trigger intracellular signaling cascades that affect the behavior of cardiac ion channels [32]. In detail, stimulation of the CB1 receptor has been shown to inhibit L-type calcium channels via Gi/o-protein-coupled receptors (GPCRs), reducing intracellular Ca^2+^ influx and affecting myocardial contractility [8,33]. Moreover, cannabinoids affect delayed rectifier potassium currents, which are essential for repolarization [16]. These effects depend on dosage and can differ based on tissue distribution and receptor subtype affinity. In addition to receptor-mediated mechanisms, some cannabinoids—especially synthetic analogs—can directly interact with ion channel proteins, modifying gating kinetics and current flow amplitude [34]. These interactions can prolong action potential duration and increase dispersion of repolarization, thereby lowering the threshold for arrhythmogenesis.

Many mechanistic pathways have been proposed to explain how cannabinoid exposure leads to SCD:Cannabinoids can affect both central and peripheral autonomic pathways, potentially causing changes that elevate the risk of arrhythmias [35].Cannabinoids may directly interact with ion channels, in addition to receptor-mediated effects, which could broaden their influence on current flow and extend repolarization phases [8].Extended exposure to cannabinoids can cause mild inflammation and oxidative stress, which may eventually result in changes to the structure and function of the myocardium [28].In individuals with pre-existing mutations, the extra stress caused by cannabinoid-related changes might reveal hidden arrhythmogenic substrates, increasing the risk of SCD [6].

To bring these different pathways together, the author of this review has created a schematic overview (Figure 2) that maps how cannabinoid-induced autonomic dysregulation, direct ion-channel modulation, oxidative stress, and unmasked genetic susceptibilities converge to lower the threshold for malignant arrhythmias and precipitate sudden cardiac death in susceptible individuals.

### 1.2. Cannabinoids in Sudden Cardiac Death

Addressing cannabinoids as a potential precipitant of SCD represents a paradigm shift in preventive cardiology. In the past, cannabis was regarded as a largely benign recreational drug of abuse; however, contemporary formulations—particularly synthetic cannabinoids—have shown proarrhythmic and vasospastic effects capable of precipitating fatal outcomes [3,18,36]. It is well known that even edible formulations have demonstrated effects comparable to those of inhaled routes on endothelial impairment, challenging the myth of their safety [37]. Furthermore, in perioperative settings, cannabis use correlates with greater susceptibility to arrhythmias and adverse drug interactions [38].

The increasing global legalization and normalization of cannabis use has prompted renewed attention on its potential health implications—particularly concerning the cardiovascular system. Though the potential benefits of cannabinoids are extensively studied for conditions like chronic pain, epilepsy, and nausea, concerns have been raised about their link to SCD, and the evidence to support that link is growing, with an obvious elevation in cardiovascular risk being noted among cannabis users. More precisely, a twofold increase in cardiovascular mortality was reported by Storck et al. (2025) [39], while Silver & Glantz (2025) reported a 29% rise in acute coronary syndrome linked to cannabis exposure and a 20% increase in stroke risk [40]. These figures are especially significant given the increasing potency and accessibility of cannabis products.

The pathophysiological mechanisms involved in the activation of CB1 receptors can lead to adverse effects such as sympathetic stimulation, endothelial dysfunction, and QT interval prolongation [41]. Early-onset atrial fibrillation, bundle branch blocks, and ventricular arrhythmias are linked to cannabis use according to Holt et al. (2024) [35]. Several-fold-more-potent synthetic cannabinoids are frequently associated with forensic cases of unexplained SCD among young individuals [18,36]. A growing number of autopsy reports identify cannabinoids in toxicological analyses of individuals who died suddenly without prior cardiac history. This suggests not only a possible temporal link to consumption but also a silent epidemic of underreported cardiovascular deaths. It is precisely among younger populations that daily use has been linked to a sixfold increase in myocardial infarction risk [26], with peak cardiovascular effects often occurring within one hour of use [42]. These acute presentations challenge conventional screening methods and highlight the need for clinicians to inquire about cannabinoid exposure in cardiovascular evaluations.

Despite convincing evidence, these cases remain unresolved, public awareness remains low, and healthcare guidelines are infrequent in addressing cannabinoid-related cardiovascular risk. Experts advocate for a regulatory approach similar to that of tobacco—complete with packaging warnings, public education campaigns, and protection from secondhand exposure [40]. Screening for cannabis use should become a standard part of cardiovascular risk assessment once an appropriate personal risk assessment tool is accepted (the protocol drafted in Figure 3 may be used as a paragon).

The proposed protocol allows data on co-intoxicants, toxicology findings, and synthetic cannabinoid potency to reinforce the forensic and mechanistic arguments already being made. The term “polypharmacy” can be used for convenience, referring to the presence of more than one drug, which was found in 61% of those with positive toxicology [43]. Published SCD cases associated with cannabinoid use have covered the following, highlighting toxicological results and poly-substance abuse patterns:Synthetic cannabinoid predominance and potency—In 278 published human cases of synthetic cannabinoid exposure, 64 different Synthetic Cannabinoid Receptor Agonists (SCRAs) have been identified [44]. These compounds are often full agonists of CB1 receptors, with far higher potency than THC [45], leading to unpredictable cardiac effects (e.g., extremes of brady or tachycardia, autonomic lability). Fatalities cluster around the more potent SCRAs (e.g., MDMB-CHMICA, 5F-MDMB-PICA), reflecting their steep dose–response curves and narrow safety margins [20,44,46].A few years ago, in a Danish nationwide study of individuals aged 1–49 who died suddenly, it was found that 77% of medico-legal autopsies on SCD cases included toxicological analysis [43]. Of these, 57% tested positive for at least one substance, either licit or illicit. The most commonly detected substances were psychotropic drugs, identified in 62% of cases, mostly at therapeutic or subtherapeutic levels. Notably, cases with positive toxicology showed a higher incidence of sudden arrhythmic death syndrome (SADS) compared to those with negative toxicology—56% versus 42%—implying that these substances may contribute to arrhythmia risk even when not at lethal concentrations [47,48].To sharpen causal inferences in sudden death cases, a separate study combined toxicological, clinical, autopsy, and genetic data, particularly in young and middle-aged victims. This comprehensive approach aids in determining if substances played a role in the death or were incidental [49].

## 2. Aim and Hypotheses

This review reanalyzes the evidence on cannabinoid exposure and its link to cardiac arrhythmias and SCD, concentrating on how inherited ion channel mutations and genetic variants influence individual risk. It also explores the molecular and electrophysiological mechanisms by which cannabinoids interfere with cardiac conduction, reveal hidden channelopathies, and trigger fatal arrhythmias.

Exposure to cannabinoids can disturb electrophysiological balance, revealing hidden genetic channelopathies and reducing the threshold for arrhythmias. Specifically, individuals with mutations in critical genes (such as *SCN5A*, *KCNH2*, *KCNQ1*, *RYR2*, and *NOS1AP*) are at higher risk of fatal cardiac events when exposed to cannabinoids. Integrating genetic screening with information on cannabinoid exposure can enhance personalized risk evaluations and strengthen cardiac safety protocols [50].

## 3. Evidence and Clinical Observations

Epidemiological studies, though still initial, indicate possible connections between cannabinoid use and adverse heart-related outcomes, such as SCD. However, confounding factors such as polypharmacy, lifestyle habits, and existing health conditions make it challenging to establish a direct cause-and-effect relationship.

In many areas, SCD is increasingly becoming a larger part of all cardiovascular deaths. Differences in emergency medical services attendance, autopsy rates, and case definitions complicate comparisons across populations. An important question is therefore raised: Should regulatory agencies implement mandatory cardiovascular warnings on cannabinoid products [51]? In Western populations, SCD accounts for approximately 15–20 percent of all deaths from cardiovascular disease, imposing a heavy toll on healthcare systems and families alike [52,53]. Community-based studies report annual SCD rates of 40–100 per 100,000 person-years in the general adult population, with the highest incidence after age 45 and a fourfold greater risk in middle-aged men. In individuals under 35 years, SCD is much rarer—approximately 1–3 per 100,000 per year—and is more often linked to cardiomyopathies, congenital anomalies, or channelopathies rather than coronary artery disease [54,55].

Although mortality from coronary heart disease has significantly decreased in recent years, the decrease in out-of-hospital SCD has been less pronounced. The rise in case reports of sudden cardiac events associated with cannabinoid use underscores the need for careful investigation. Although cannabinoids have therapeutic benefits, their impact on cardiac electrophysiology in vulnerable groups must be considered. Healthcare providers, researchers, and policymakers should acknowledge these risks without inducing undue alarm. These concerns underscore the importance of standardized monitoring, improved risk assessment, and targeted prevention strategies to reduce the global incidence of SCD.

The exact mechanism that steers cannabinoid-exposed individuals to SCD, especially if they have genetic risks, requires more detailed study. While official guidelines on genetic testing are still being developed, education for healthcare professionals and the public is crucial to minimizing dangers. Furthermore, adding genetic risk assessments in postmortem healthcare, expanding epidemiological surveillance, and creating targeted educational initiatives are essential steps toward safer cannabinoid use. As ongoing research continues to uncover this complex relationship, personalized medicine may become an effective means of balancing treatment benefits with cardiovascular safety [56,57].

### 3.1. Aggregated Data on Cannabinoids and Sudden Cardiac Death

Most up-to-date observational and meta-analytic evidence linking cannabis use to cardiovascular outcomes, with a focus on SCD, is presented below, and it consistently shows a two- to threefold increase in cardiovascular mortality among cannabis users [26,39,58].

SCD is categorized within the class of “cardiovascular death,” and there are no specific studies focused solely on SCD. Most evidence comes from observational studies that may be affected by confounding factors, such as tobacco and poly-substance use.

Many case reports associate acute cannabis intoxication with SCD, including one documented incident of accidental death where heavy cannabis use preceded fatal arrhythmia [59]. These anecdotal data highlight plausibility but lack the power to establish causality or quantify population-level risk [60,61].

A pooled analysis of 24 observational studies (≈200 million participants), published online in the journal Heart, found the following: acute coronary syndrome (ACS): relative risk (RR) 1.29 (95% confidence interval (CI) = 1.05 to 1.59); stroke: RR 1.20 (1.13 to 1.26); cardiovascular death (including SCD): RR 2.10 (1.29 to 3.42); no significant association for the composite ACS + stroke outcome [39].

Using Medical Xpress, the same 24 studies were analyzed, and the results reinforced cardiovascular mortality findings: 29% higher risk of ACS, 20% higher risk of stroke, 100% higher risk (2×) of cardiovascular death. Authors advise caution regarding the presence of moderate-to-high bias due to exposure misclassification and observational designs, but note consistent doubling in cardiovascular (CV) death risk [62].

Among 4.6 million adults under 50 (TriNetX, ACC.25) without baseline cardiometabolic conditions, cannabis users experienced over a sixfold increase in myocardial infarction risk, a fourfold increase in ischemic stroke risk, double the risk of heart failure, and three times the risk of a composite outcome (including cardiovascular death, MI, or stroke). The average follow-up period was three years [26].

Researchers at the American College of Cardiology’s 74th Annual Scientific Session & Expo pooled data from 12 studies (n ≈ 75 million) and observed a 50% increased risk of heart attack among cannabis users, a feature consistent across the U.S., Canada, and India. The study had moderate to good quality, but its findings were limited by heterogeneity and its observational design [63]. Key observational studies included in this review are outlined in Table 2.

Future research in this area should focus on isolating SCD events. Incorporating data from registries—like national cardiac arrest databases—or autopsy series may lead to an improvement. Future work should also aim to standardize measures of cannabinoid exposure and examine dose–response relationships. Beyond mortality, it is essential to investigate arrhythmogenic mechanisms such as cannabinoid-induced tachycardia, QT prolongation, and coronary vasospasm to better understand how cannabinoids might trigger sudden lethal events.

### 3.2. Aggregated Data on Poly-Substance Exposure and Sudden Cardiac Death

Some cases show that cannabis—when detected in SCD victims—is part of a broader poly-substance pattern, underscoring the need to perform broad toxicological assessments to evaluate combined cardiotoxic interactions [65]. The combined use of alcohol and cannabinoids—especially synthetic types—has been increasingly associated with serious heart problems, including SCD, making it a growing concern.

Forensic reports have identified poly-substance abuse as a factor in SCD since the beginning of comprehensive toxicological screening. In 2011, Ceelen et al. [66] analyzed over 200 medico-legal cases and found that cannabis metabolites were present alongside two to three other substances in approximately 40% of SCDs. Regarding the presence of synthetic cannabinoids, a study by Giorgetti et al. highlighted the challenge in identifying their toxic or lethal doses, given that many cases have exhibited adverse cardiac effects such as arrhythmias and myocardial infarction, especially when combined with alcohol [19]. Lynge et al. [43] reported in 2018 that at least one drug was detected among 1–49-year-olds, representing 61% of the SCD cohort; cannabis was found in 11% of poly-substance screens. Drummer et al. (2019) [7] described 13 SCDs attributed, at least in part, to cannabis. THC was almost always found with at least one other CNS-active drug, such as antidepressants, opioids, and alcohol. Among 278 SCRA cases, there were 840 total drug co-exposures—an average of four substances per case [44]. The most common co-intoxicants were alcohol (11.4%), opioids (11.2%), and cannabis itself (11.1%). Mortality was significantly higher when SCRAs were combined with antipsychotics/antidepressants, alcohol, or tobacco (*p* < 0.05), suggesting additive or synergistic cardiotoxic interactions [44]. Regarding ongoing research on synthetic cannabinoids, a 2025 animal study on mice found that binge drinking combined with synthetic cannabinoids like CP55,940 caused greater cardiac depression than using either substance individually [67]. The heart’s contractile function was notably reduced, and both central and peripheral CB1 receptor signaling pathways were involved [65].

An MDMB-CHMICA-related SCD (22-year-old male found collapsed approximately 15 min after inhaling a brown powder substance) was considered as a case example [46]. Toxicology: MDMB-CHMICA detected in blood (concentration not reported), no other drugs found. Outcome: asystole on EMS arrival; later died of hypoxic brain injury despite resuscitation. Key point: a single exposure to a high-potency SCRA can be fatal by triggering malignant arrhythmia in an otherwise healthy heart. Another case example concerned 5F-MDMB-PICA and 4F-MDMB-BINACA fatality (33-year-old male) [68]. In this case, blood toxicology showed 5F-MDMB-PICA 0.9 ng/mL, and urine analysis showed 0.1 ng/mL, cerebrospinal fluid (CSF) 3.2 ng/mL; 4F-MDMB-BINACA was found only in CSF (0.1 ng/mL). No other co-ingestants were reported; however, it is worth noting that the rapid elimination of parent substances highlights the importance of metabolite screening in postmortem workups. A systematic review published in July 2025 highlighted synthetic cannabinoids as one of the most frequently reported substances in sudden death cases [18]. Young adult males were disproportionately affected, and postmortem toxicology often revealed synthetic cannabinoids alongside alcohol or other substances.

Very recently, a case report detailed a patient who suffered multiple cardiac arrests due to coronary vasospasm. The combination of cannabinoids and caffeine initiated the vasospasm. The patient’s history also included alcohol use, suggesting a possible synergistic risk factor that could enhance the “individual risk assessment tool” to be developed [69].

### 3.3. Limitations of the Evidence Base

Despite the accumulating body of epidemiological and forensic evidence, the studies considered exhibit several significant limitations. Most of the data derive from observational designs and case series, which are inherently subject to residual confounding and cannot establish causality [7,39]. Exposure assessment varies widely—from self-reported cannabis use in large cohorts [26,35] to post-mortem toxicological screens with inconsistent autopsy rates [43,47,54]—leading to potential misclassification and underestimation of true prevalence. Meta-analyses and pooled real-world analyses often combine heterogeneous endpoints (e.g., acute coronary syndrome, stroke, and SCD) and employ different definitions of cardiovascular death, which may inflate or obscure specific risks [59,62]. Furthermore, many mechanistic insights are restricted because there are case reports of synthetic cannabinoid fatalities [46] that lack standardized dose–response data and comprehensive genetic autopsy to distinguish incidental findings from causative arrhythmogenic events [49,70]. The limited number of prospective cohort studies that include both molecular autopsy and genetic profiling restricts our capacity to accurately measure the interaction between cannabinoid exposure and hidden channelopathies.

### 3.4. Mechanisms by Which Poly-Drug Use Amplifies Risk

Alcohol and psychotropics can extend cardiac repolarization [71,72], increasing the risk of torsades de pointes when combined with SCRAs that affect calcium and potassium currents [8,73]. Opioids and even nicotine disturb autonomic balance further, intensifying SCRA-induced sympathetic surges and vagal response swings [74]. Many prescription drugs, such as some antipsychotics and antidepressants, inhibit cytochrome P450 (CYP) enzymes responsible for metabolizing SCRAs [75]. This inhibition can increase circulating SCRA levels and enhance CB1-mediated cardio depressant or proarrhythmic effects [21,76]. Activation of CB1 receptors, particularly by synthetic cannabinoids, can damage heart function [77]. Alcohol and cannabinoids can enhance each other’s impact on the cardiovascular system, causing arrhythmias, decreased cardiac output, and blood pressure fluctuations [67]. Young males with histories of substance use are particularly vulnerable, often presenting with no prior cardiac symptoms before fatal events. However, blocking these receptors has demonstrated potential in reversing certain damage in animal studies, a feature that should be employed in developing targeted therapeutics [12].

The key mechanisms can be described as follows:When two or more agents prolong ventricular repolarization or depress myocardial excitability, their effects combine or even enhance each other, a phenomenon known as the pharmacodynamic combined effect. This effect leads to lowering the threshold for torsades de pointes and other malignant rhythms [78].Co-ingested drugs can inhibit or compete for the same cytochrome P450 enzymes, causing one or more substances (e.g., SCRAs) to accumulate to toxic levels and extend their cardiac effects [79].Alcohol, opioids, nicotine, and psychotropics each disturb the sympathetic–parasympathetic equilibrium; when combined with SCRAs, changes in heart rate and blood pressure become more extreme, triggering electrical instability [80,81].Increased oxidative stress and inflammation refer to the presence of multiple xenobiotics, which elevate reactive oxygen species and induce low-grade inflammation in the myocardium, further impairing structural and cellular damage and disrupting ion channel function and conduction homogeneity [82,83].Additive hemodynamic effects may be noticed when several central nervous system depressants or stimulants are used together, causing alternating episodes of hypotension, hypertension, tachycardia, or bradycardia, with each one recognized as an arrhythmic trigger when the heart’s compensatory reserve is overwhelmed [84].

## 4. Interplay Between Genetics and Cannabinoid Exposure

The clinical ramifications of this interplay are significant [85]. For genetically predisposed individuals, the additional electrophysiological stress introduced by cannabinoids may reduce the threshold for life-threatening arrhythmias. For instance, a slight decrease in repolarization reserve (the myocardium’s built-in safety margin for restoring the action potential’s repolarization phase) caused by an ion channel mutation might be worsened by cannabinoid-induced changes in calcium regulation or autonomic nervous system activity [8]. This scenario creates a “shipwreck”, as a genetic defect combined with drug therapy can lead to events like ventricular tachycardia or fibrillation.

From a clinical perspective, such insights argue for a more personalized approach. Genetic screening for individuals with a family history of SCD or known channelopathies can help identify those at higher risk when exposed to cannabinoids. This allows healthcare providers to give personalized guidance on using cannabinoid products, whether for medical or recreational use, and may also promote closer monitoring of arrhythmic events in this group.

### 4.1. Cardiac Electrophysiology and Sudden Cardiac Death

Sudden cardiac death is often the catastrophic endpoint of complex arrhythmogenic processes. A delicate balance among ion channel functions, cellular connectivity, and autonomic input sustains the heart’s rhythm. Disruption of these components—whether via ischemia, structural heart disease, or disturbances in ion channels—can precipitate fatal arrhythmias. Research has increasingly identified several triggers and substrates for SCD, highlighting the significance of both external factors (such as drugs or toxins) and inherent genetic component vulnerabilities.

### 4.2. Genetic Variants Implicated in Cardiac Arrhythmias

Based on current genetic research, the following five single-nucleotide polymorphisms (SNPs) have been identified as associated with an increased risk of SCD [36]. Some are common variants with modest effect sizes (odds ratio (OR) > 1, indicating an increased risk, or <1 indicates a protective effect), while others are rare but highly penetrant. These SNPs might not cause disease on their own, but can reduce the threshold for arrhythmia when combined with environmental triggers like cannabinoid exposure. They are located in genes that encode ion channels or their modulators related to cardiac electrophysiology, as summarized in Table 3.

Cannabinoids, which have complex effects on the cardiovascular system, might act as such triggers, revealing these hidden susceptibilities [10]. This raises the question: Should individuals with a genetic predisposition to arrhythmias undergo screening before using cannabinoids? As personalized medicine rapidly develops, incorporating genetic risk evaluations into standard health assessments could help identify people at greater risk of cannabinoid-related arrhythmias.

Numerous genetic variants that influence cardiac ion channel activity and repolarization dynamics have been identified. Mutations in genes encoding sodium, potassium, and calcium channels can predispose individuals to conditions such as long QT syndrome, Brugada syndrome, and catecholaminergic polymorphic ventricular tachycardia. These variants may remain clinically silent until environmental or pharmacological triggers unmask them. The presence of such variants compromises the cardiac conduction reserve and heightens the risk of arrhythmia under stress [8,14].

Genetic predisposition has a significant impact on the cardiovascular response to cannabinoids [10]. Individuals with specific ion channel mutations might have a reduced capacity to handle unexpected disruptions in electrophysiological balance. When cannabinoids either further modulate ion channel function or induce autonomic imbalance (disturbance), the likelihood of triggering malignant arrhythmias significantly increases [34]. This combined effect highlights the importance of genetic screening and personalized guidance on cannabinoid use, especially for at-risk groups.

Although no current tool exists to assess the genetic risk of SCD in cannabis users, a conceptual framework for developing such a tool should include the components listed in Table 4. Since the ultimate goal of assessing genetic risk is to create a comprehensive tool for evaluating individual risk, genotyping panels targeting cardiac arrhythmia genes should be paired with a user questionnaire that includes behavioral history and a cannabis potency index (e.g., THC %).

### 4.3. Channelopathies, Their Genetic Background, and Relation to Cannabinoids

The genetic basis of channelopathies is well established. For example, mutations in the *SCN5A* gene, which encodes the cardiac sodium channel, are commonly associated with Brugada syndrome and specific variants of long QT syndrome [14]. Likewise, some genes involved in potassium channels, such as *KCNH2* and *KCNQ1*, have been linked to long QT syndrome [15]. In contrast, alterations in the ryanodine receptor gene (*RYR2*) often underpin catecholaminergic polymorphic ventricular tachycardia [92]. These mutations alter the normal function of ion channels—in some cases, by changing the gating properties or the conductance of the ion channels—thus disrupting the delicate balance of ionic currents during the cardiac action potential, and predisposing individuals to arrhythmias and, in some cases, SCD.

Channelopathies are genetic disorders resulting from mutations in genes that encode ion channels, the protein complexes that facilitate the movement of ions, such as sodium, potassium, and calcium, across cell membranes [8,14]. At the core, these channels are essential for starting and regulating electrical signals that govern the heartbeat rhythm. Notable cardiac channelopathies involve conditions such as long QT syndrome, Brugada syndrome, and catecholaminergic polymorphic ventricular tachycardia. These genetic conditions may remain hidden for years until an external trigger, such as a fever, medication, or metabolic disturbance, triggers arrhythmic episodes. In that sense, the use of cannabinoids has emerged as a possible triggering factor for these electrical activities [14].

For this review, a hypothetical dataset was generated, consisting of ten individuals with a polygenic risk score (PRS) ranging from 0.12 to 1.38 and their corresponding estimated cardiac risk. To illustrate how genetic burden maps onto predicted cardiac risk, the PRS for each individual was plotted against their estimated risk (Figure 1). As shown in Figure 4, risk grows linearly with PRS (y = 5.128x − 122.54; R^2^ = 0.94), with high-risk genotypes clustering in the right-hand tail.

## 5. Illustrative Model

To test hypotheses, a model of 10 SCD cases featuring anonymized toxicological screening data from postmortem urine specimens containing identified cannabinoids was analyzed. Although real forensic case data are not typically shared in raw form, the values and trends presented here are based on concentration ranges and patterns reported in the forensic toxicology literature, specifically in the work of Ceelen et al. (2010) [66] and Lemos et al. (2011) [17]. The primary cannabinoid in these cases was the metabolite THC-COOH, with the occasional detection of delta-9-THC itself and no detection of 11-hydroxy-THC, which is in line with published observations.

All publicly reported SCDs and severe cardiovascular emergencies that occurred close in time to cannabinoid use were mainly sourced from Drummer et al.’s forensic review (2019) and supplemented by clinical observations in the literature. They were classified based on whether they were case-fatal SCDs (n = 13). For the 13 cases, ages ranged from 19 to 65 years (mean = 34 years) [7]. Moreover, the review identified 35 non-fatal cardiovascular emergencies (n = 35) [93]. These included hospital presentations for arrhythmias, myocardial infarction, stroke, and cardiovascular events within hours of cannabis use. Table 5 shows the age distribution of fatal vs. non-fatal events, highlighting a predominance among young adults.

Young adults (<30) constitute most fatal and non-fatal events, middle-aged users (30–50) account for roughly a quarter of SCDs but nearly a third of emergencies, and older individuals (>50) are underrepresented in case reports, though existing data likely underestimate their risk, especially given age-related cardiovascular vulnerability. These data differ significantly from those presented by Lemos et al. (2011) [17] or those by Ceelen et al. (2010) [66].

## 6. Clinical Implications and Future Directions

Both medical and recreational use of cannabinoids has increased rapidly; however, cannabinoids’ impact on cardiovascular health remains a concern, particularly for individuals with genetic predispositions [10]. This review outlines essential tracks by which cannabinoids could increase the risk of arrhythmias and SCD, focusing on ion channel effects, autonomic nervous system disturbances, and oxidative stress.

The clinical significance of genetic susceptibility to SCD related to cannabis is increasingly recognized in cardiology, toxicology, and public health. Individuals with harmful variants in genes such as *SCN5A*, *KCNQ1*, *KCNH2*, *RYR2*, and *NOS1AP* [88,89,90], as well as other genes coding for structural or regulatory proteins, including *CALM2*, *PKP2*, *MYH7*, *TNNI3*, and *MYBPC3* [94,95], *may* have a reduced repolarization reserve [16]. Indeed, cannabis activates CB1 receptors in the heart, potentially leading to autonomic imbalance, oxidative stress, and changes in ion channels. Especially at high THC levels, cannabis can act as a trigger by exposing hidden channelopathies and causing arrhythmias [6]. These effects are worsened in individuals with genetic predispositions, creating a “shipwreck” for arrhythmias. Understanding how cannabinoids interact with genetic variations in SCD is crucial for effective clinical practice. As a first step, genetic screening could be necessary before prescribing cannabinoids, particularly for patients with a family history of arrhythmias or unexplained syncope. Routine ECG monitoring may also be recommended within this group.

The link between cannabinoids, genetics, and SCD presents a complex challenge that requires thorough investigation and thoughtful analysis [85]. Since cannabis use roughly doubles the risk of cardiovascular death, including stroke and heart attack [96], it is not surprising that regulatory agencies might require warnings about cardiovascular risks on cannabis products [26]. Accordingly, clinicians should inquire about cannabis use during cardiovascular risk assessments, especially in younger patients without traditional risk factors. Healthcare providers ought to factor in a patient’s genetic background when offering guidance on cannabinoid use, particularly in groups with a history of arrhythmias or unexplained cardiac issues. Cannabis should therefore be considered a serious cardiovascular risk factor—similar to tobacco or alcohol—when evaluating patients [91].

In cases of SCD, especially among young adults, postmortem toxicology should include cannabinoid screening. An additional genetic autopsy (molecular autopsy) can help differentiate between incidental cannabis use and causative arrhythmogenic death [5,6,70]. Such a detailed approach could eventually support tailored public health guidelines and regulatory policies concerning cannabinoid consumption.

Another critical discussion point is the consideration of genetic variants associated with arrhythmias and how cannabinoids might act as environmental triggers, potentially exacerbating hidden cardiac risks [5,6]. Aside from that, future research directions need to be defined. The area where cannabinoid pharmacology intersects with genetic predisposition is still largely unexplored, and additional studies are needed [97,98]. Animal models and in vitro experiments offer insights into how cannabinoids influence ion channels [13,99,100]. Long-term cohort studies incorporating genetic profiling could also help evaluate real-world risks [101].

### Future Directions and Research Challenges

Personalized medicine is advancing rapidly, and integrating genetic risk assessments into routine medical evaluations could help identify individuals at higher risk for cannabinoid-induced arrhythmias. The intersection of cannabinoid pharmacology, genetic factors, and heart electrophysiology presents a valuable avenue for upcoming research. Major challenges are anticipated to include the following:Additional animal and cell-based research is required to clearly understand the molecular interactions between cannabinoids and different ion channels.In addition, observational studies with genetic screening can help measure the rate of SCD among cannabinoid users and identify groups at higher risk.Likewise, trials in progress should be carefully designed to evaluate interventions or guidelines that can mitigate arrhythmogenic risk in individuals who are genetically predisposed.Personal risk assessment tools must be developed and activated, and their implementation clearly defined. Tools may include genetic testing, targeted ECG monitoring, and accurate dosing strategies to minimize risks while maintaining therapeutic effectsProgress in fields related to genomics (e.g., bioinformatics) could eventually lead to personalized cannabinoid use for treatment, optimizing the benefits and reducing the adverse cardiac effects of such treatment.

## 7. Conclusions

Although case series and retrospective cohorts suggest a link from cannabinoid exposure to SCD in carriers of channelopathy-associated variants, the evidence remains limited by heterogeneous endpoints and lack of dose–response data. While therapeutic promise drives wider cannabinoid use, observational designs and inconsistent toxicology thresholds prevent firm conclusions about arrhythmic risk—underscoring the need for prospective safety trials. A systematic, prospective evaluation of genotype–exposure interactions—ideally in ECG-monitored, genotype-stratified cohorts—is essential to translate these mechanistic insights into personalized prescribing guidelines. Moving beyond in vitro and postmortem case reports, multicenter cohort studies with standardized cannabinoid dosing, serial ECG/biomarker monitoring, and molecular autopsy panels are required to establish causality.

Future pharmacogenomic trials should quantify how metabolic SNPs (e.g., CYP2C9, CYP3A4 variants) modulate plasma THC/CBD curves and downstream electrophysiological effects on susceptible genotypes. Dedicated drug-interaction studies are needed to measure how common co-medications (antidepressants, antipsychotics, alcohol) alter cannabinoid PK/PD and arrhythmia thresholds in genetically predisposed patients. In parallel with evolving regulations, a pilot implementation of a genetic-ECG screening protocol (e.g., SCN5A/RYR2 panel + baseline QTc) in medical cannabis clinics should be evaluated for feasibility and predictive value.

## Figures and Tables

**Figure 1 ijms-26-08635-f001:**
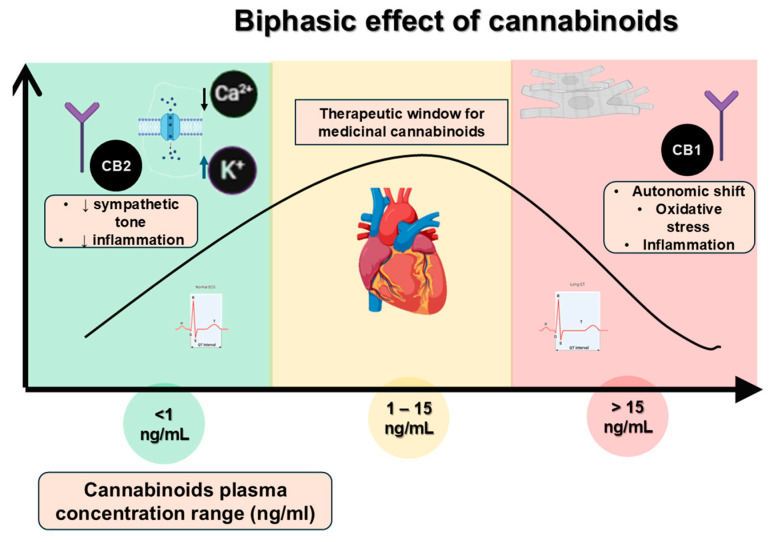
Biphasic effects of cannabinoids on cardiac electrophysiology. Low doses may exert cardioprotective effects through CB1/CB2 receptor activation and modulation of Ca^2+^/K^+^ currents, whereas high doses can increase arrhythmogenic risk via sympathetic overstimulation and ion channel dysregulation. (Abbreviations: CB1, cannabinoid receptor type 1; CB2, cannabinoid receptor type 2; ANS, autonomic nervous system; Ca^2+^, calcium ion; K^+^, potassium ion).

**Figure 2 ijms-26-08635-f002:**
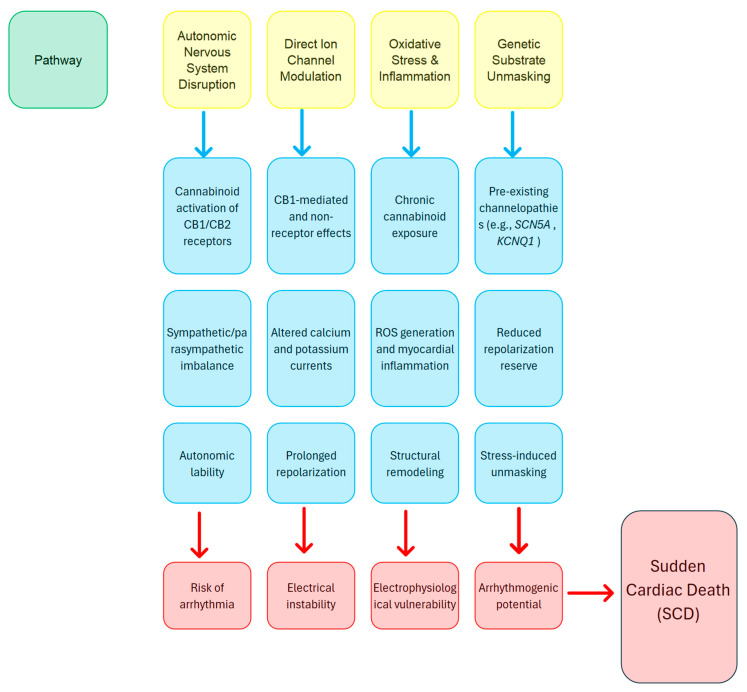
Proposed mechanistic pathways linking cannabinoid exposure to sudden cardiac death (SCD). Cannabinoid-induced autonomic dysregulation, ion channel modulation, oxidative stress, and genetic unmasking converge to increase arrhythmogenic potential. These mechanisms may act synergistically or independently to precipitate lethal cardiac events, particularly in genetically predisposed individuals. (Abbreviations: CB1, cannabinoid receptor type 1; CB2, cannabinoid receptor type 2; ANS, autonomic nervous system; Ca^2+^, calcium ion; K^+^, potassium ion; ROS, reactive oxygen species; SCD, sudden cardiac death).

**Figure 3 ijms-26-08635-f003:**
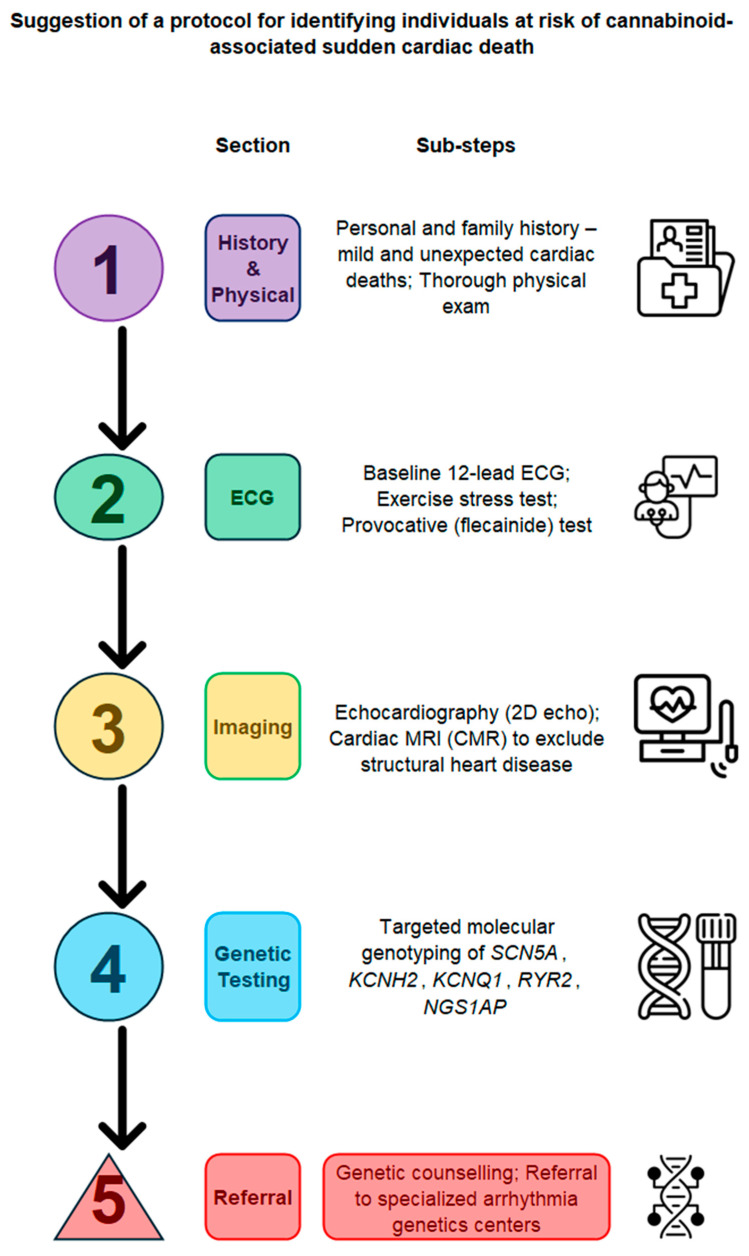
Suggestion of a protocol for identifying individuals at risk of cannabinoid-associated sudden cardiac death. This flowchart outlines a four-step screening pathway. (Abbreviations: SCD, sudden cardiac death; ECG, electrocardiogram; QTc, corrected QT interval; SCRA, synthetic cannabinoid receptor agonist; CYP, cytochrome P450; UDS, urine drug screen).

**Figure 4 ijms-26-08635-f004:**
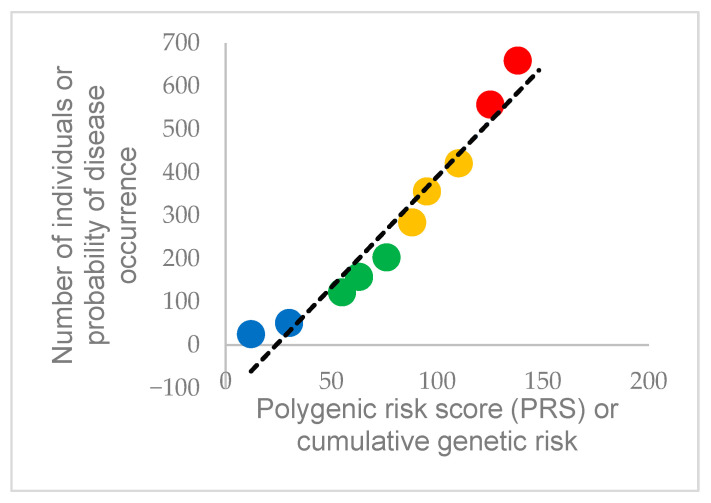
Scatterplot of polygenic risk score (PRS) versus estimated cardiac risk percentage. Data points are color-coded by risk group (Low: blue; Moderate: green; High: orange; Critical: red). The fitted linear trendline (y = 5.128x − 122.54; R^2^ = 0.94) highlights the strong association between genetic predisposition and projected cardiac risk. The dataset was generated using Copilot, Version 1.0; Microsoft Corporation: Redmond, WA, USA, 2025; Available online: https://copilot.microsoft.com (accessed on 23 July 2025). The author has reviewed and edited the dataset, performing additional calculations using Microsoft Office 365 by Microsoft Corporation: Redmond, WA, USA, 2025; Available online: https://www.microsoft.com/en-us/microsoft-365 (accessed on 23 July 2025). The Copilot-generated dataset is provided in Appendix A. A third data series—identical to the preceding two—was intentionally included to avoid system hallucinations.

**Table 1 ijms-26-08635-t001:** Concentration-dependent effects of cannabinoids.

Cannabinoid	Plasma Concentration Range (ng/mL)	Observed Effects	References
Δ^9^-THC (low dose)	1–5	Sedation, anti-inflammatory benefits	[24]
Δ^9^-THC (moderate dose)	5–15	Mild tachycardia, autonomic fluctuation	[25]
Δ^9^-THC (high dose)	>15	Marked tachycardia, hypertension, autonomic instability	[26]
Synthetic cannabinoids (low dose)	<1	Anxiety, mild autonomic symptoms	[27]
Synthetic cannabinoids (high dose)	≥1	Severe autonomic dysregulation, malignant arrhythmias	[27]
Endocannabinoids (low dose)	<2	Homeostatic regulation, mild anti-inflammatory effects	[9]
Endocannabinoids (high dose)	≥2	Dysregulation of autonomic tone, pro-arrhythmic potential	[8]

Abbreviations: Δ9-THC, Δ9-tetrahydrocannabinol; ng, nanogram; mL, milliliter.

**Table 2 ijms-26-08635-t002:** Summary of key observational and meta-analytic studies linking cannabis use to cardiovascular outcomes, including sudden cardiac death (SCD).

Study/Source	Design	n (Approx.)	Outcome	Effect Size	Citation
Injury and Death—NCBI case report	Case report	1	SCD	Anecdotal	[64]
Storck et al. (Heart, 2025)	Systematic review + meta-analysis	200 million	Cardiovascular death	RR 2.10 (1.29–3.42)	[39]
Medical Xpress pooled analysis (June 2025)	Pooled epidemiology	200 million	Cardiovascular death	2× risk	[62]
TriNetX retrospective (ACC.25)	Cohort, follow-up 3 y	4.6 million	Composite CV death/MI/stroke	3× risk	[26]
ACC meta-analysis (12 studies)	Meta-analysis	75 million	Myocardial infarction (heart attack)	OR 1.50	[63]

Abbreviations: SCD, sudden cardiac death; CV, cardiovascular; MI, myocardial infarction; RR, relative risk; OR, odds ratio.

**Table 3 ijms-26-08635-t003:** Single-nucleotide polymorphisms (SNPs) linked to SCD risk. For each variant, it lists the gene name, the ID of the SNP, the effect size (odds ratio, OR), the type of alteration, the geographic populations in which it is most frequent, and a brief note on its functional or clinical significance.

Gene	SNP ID	Effect Size (Odds Ratio, OR)	Variant Type	High-Frequency Regions	Notes
** *SCN5A* **	rs11720524	0.76	Common and rare SNPs	East Asia, Europe, South Asia	Associated with reduced SCD risk in Europeans [18,86]
** *KCNQ1* **	rs2283222	0.73	SNPs and deletions	Japan, Korea, Northern Europe	Linked to QT interval modulation and lower SCD risk [18,86]
** *KCNQ1* **	rs12296050	0.85	SNPs and missense	Europe, North America	Protective effect observed in Koreans and Americans [18,86]
** *RYR2* **	rs790896	0.66	Missense mutations	Japan, Italy, USA	Associated with reduced risk of catecholaminergic polymorphic VT [18,86]
** *NOS1AP* **	rs16847548	1.28	Common SNPs	Europe, North America	Increases QT interval and risk of cardiac events, including SCD [18,87]

Abbreviations: SNP: single-nucleotide polymorphism; SCD: sudden cardiac death; OR: odds ratio; VT: ventricular tachycardia; *SCN5A*: sodium voltage-gated channel alpha subunit 5; *KCNQ1*: potassium voltage-gated channel subfamily Q member 1; *RYR2*: ryanodine receptor 2; *NOS1AP*: nitric oxide synthase 1 adaptor protein.

**Table 4 ijms-26-08635-t004:** Components of the proposed genetic risk assessment tool for sudden cardiac death in cannabis users.

Component	Description	References
Genetic Markers	Single-nucleotide polymorphisms linked to cardiac arrhythmias (e.g., *SCN5A*, *RYR2*, *KCNQ1*, *KCNH2*, and *NOS1AP*)	[18,36,88,89,90]
Epigenetic Modifiers	DNA methylation changes triggered by cannabis or alcohol exposure	[28,34,67,76]
Cannabis Use Profile	Frequency, duration, type (THC/CBD ratio), age of initiation	[26,30,39,58]
Family History	Sudden death, inherited cardiac diseases, substance use disorders	[6,7,36,70]
Lifestyle Factors	Alcohol use, physical activity, stress, sleep quality	[71,72,74,82,83]
Heart Health Parameters	ECG abnormalities, QT interval, previous arrhythmia episodes	[5,14,35,91]
Interaction	Synergistic risk from substance interactions (e.g., alcohol + cannabinoids)	[19,44,66,67,69]

Abbreviations: SCN5A: sodium voltage-gated channel alpha subunit 5; RYR2: ryanodine receptor 2; KCNQ1: potassium voltage-gated channel subfamily Q member 1; KCNH2: potassium voltage-gated channel subfamily H member 2; NOS1AP: nitric oxide synthase 1 adaptor protein; DNA: deoxyribonucleic acid; THC: Δ^9^-tetrahydrocannabinol; CBD: cannabidiol; ECG: electrocardiogram; QT: interval between Q and T waves on an ECG.

**Table 5 ijms-26-08635-t005:** Age distribution of cannabinoid-related cardiac events [7,93].

Age Group (Years)	SCDs (n = 13)	% of SCDs	Non-Fatal Emergencies (n = 35)	% of Emergencies
<30	8	61.5%	22	62.9%
30–50	3	23.1%	11	31.4%
>50	2	15.4%	2	5.7%

Abbreviations: SCD, sudden cardiac death; n, number of cases.

## Data Availability

Available upon request to ivan.sosa@uniri.hr.

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
