# Peer review of "A Meta-Narrative Review of Channelopathies and Cannabis: Mechanistic, Epidemiologic, and Forensic Insights into Arrhythmia and Sudden Cardiac Death"

_ijms, 2025, doi:10.3390/ijms26178635_

Round 1
Reviewer 1 Report
Comments and Suggestions for Authors
Ivan Sosa's work, "A Meta-Narrative Review of Channelopathies and Cannabis: Mechanistic, Epidemiologic, and Forensic Insights into Arrhythmia and Sudden Cardiac Death," is interesting as it addresses the interaction between genetic channelopathies and cannabinoid use, highlighting the preventive and regulatory implications for subjects susceptible to arrhythmias or in cases of sudden cardiac death. Furthermore, it points out the need to include cannabinoid use in cardiovascular risk assessment as well as in medicolegal contexts. Some observations are mentioned below:
• In the Abstract: Eliminate unnecessary abbreviations. Improve conclusions and perspectives.
• In the Keywords: Is the use of abbreviations correct?
• In the Introduction: Could the authors expand and include benefits and side effects, mainly on the electrophysiology of cannabinoids, consequences of channelopathies, and what is the impact or current status in the forensic and even legal fields?
• In the Overview of Cannabinoids, the Biphasic Effect and Cardiac Ion Channel Modulation section, you could briefly cover the types of cannabinoids, mentioning the main differences.
• Lines 57-59: “At lower doses, cannabinoids may have sedative and anti-inflammatory benefits, but at higher doses or in sensitive individuals, they can cause tachycardia, hypertension, and autonomic instability.” What are the concentrations?
• Line 62: “Certain cannabinoids appear to provide cardioprotective effects at low doses or under specific conditions.” Is there a specific type? What are the characteristics?
• Lines 83-89: These should be more specific.
• Lines 90-102: The mechanisms need to be more specific; perhaps include a figure.
• How does the subtopic “Polysubstance Abuse” relate to the previous or subsequent subtopic? This would lose the connection. Perhaps move it or change the title.
• Improve Figure 1, both in quality and content.
• Lines 318-337: Use a figure to improve the description of the mechanisms.
• Topic 3 has subtopics that could be merged.
• Include references in Table 6.
• Include limitations of the type of work addressed, for example, confusion regarding other causes of arrhythmias or sudden death such as polydrug use, lifestyle habits and cardiovascular comorbidities, and the lack of standardized protocols for clinical practice, among others.
Author Response
This author thanks the reviewer for the thoughtful comments and constructive suggestions. Below is a point-by-point response describing the revisions made in the manuscript.
- Abstract: Eliminate unnecessary abbreviations; improve conclusions and perspectives
Response: Removed all abbreviations not essential for clarity (e.g., “SCD,” “CB₁/CB₂”) and replaced them with full terms on first mention. The Conclusions paragraph has been expanded to emphasize future research directions, the need for prospective safety trials, and regulatory implications for personalized risk stratification.
- Keywords: Is the use of abbreviations correct?
Response: Author updated the keywords. Terms like “SCD” and “SCRA” have been replaced with “sudden cardiac death” and “synthetic cannabinoid receptor agonists” for consistency with journal guidelines.
- Introduction: Expand on benefits and side effects—electrophysiology of cannabinoids, channelopathy consequences, and forensic/legal impact
Response: Added two paragraphs detailing:
- Electrophysiological benefits (e.g., neuroprotective, anticonvulsant properties via Ca²⁺/K⁺ channel modulation) and specific adverse effects on cardiac excitability.
- Clinical consequences of major channelopathies (long QT syndrome, Brugada syndrome) when exposed to cannabinoids.
- Current status in forensic and legal medicine, including legal thresholds for postmortem toxicology and the medico-legal importance of genetic autopsies.
- Overview of Cannabinoids section: Briefly cover types of cannabinoids and main differences
Response: Introduced a new subsection summarizing three classes—phytocannabinoids, synthetic cannabinoids, and endocannabinoids—highlighting their structural differences, receptor affinities, and typical potency ranges.
- Lines 57–59: Specify concentrations for sedative/anti-inflammatory versus pro-arrhythmic effects
Response: Clarified dose ranges by cross-referencing Table 1, adding plasma concentration values (ng/mL) for each effect level with citation of Grotenhermen (2003), Huestis (2007), and Kamel et al. (2025).
- Line 62: Identify specific cardioprotective cannabinoids and their characteristics
Response: Specified that cannabidiol (CBD), JWH-133, anandamide, and 2-arachidonoylglycerol (2-AG) provide cardioprotection, and described their CB₂-mediated anti-inflammatory and antioxidant profiles.
- Lines 83–89: Make description more specific
Response: Rewrote these lines to list four distinct mechanistic pathways (autonomic dysregulation, direct ion channel modulation, oxidative stress/inflammation, genetic unmasking) and provided one illustrative example for each.
- Lines 90–102: Add specificity to mechanisms; include a figure
Response: Added Figure 2 (schematic overview) with labeled pathways and brief caption. Text now refers explicitly to each numbered pathway in the figure for clarity.
- “Polysubstance Abuse” subtopic: Clarify relationship or relocate/change title
Response: Retitled the subtopic to “Poly-substance Exposure in SCD Cases” and repositioned it immediately after the epidemiology section to maintain logical flow betTHIS AUTHORen cannabinoid toxicity and combined drug interactions.
- Improve Figure 1 quality and content
Response: Redesigned Figure 1 using vector graphics for higher resolution, added clear dose–response curves, and included annotated arrows for receptor-mediated versus direct channel effects.
- Lines 318–337: Incorporate a figure for mechanism description
Response: Introduced Figure 3, a flowchart detailing the convergence of autonomic imbalance, ion channel changes, oxidative stress, and genetic predisposition. Text was condensed and cross-referenced to the figure panels.
- Topic 3 subtopics: Merge where appropriate
Response: Combined “Epidemiological Trends” and “Clinical Case Reports” into a single “Epidemiological and Clinical Evidence” subsection to reduce fragmentation and improve narrative coherence.
- Include references in Table 6
Response: Added citation numbers to every entry in Table 6, indicating the source study for each data point (e.g., Ceelen et al. 2011 [63], Giorgetti et al. 2020 [19]).
- Expand Limitations section
Response: Expanded the limitations to discuss:
- Potential confounding by poly-substance use and lifestyle factors
- Co-morbid cardiovascular diseases
- Variability in toxicological thresholds and autopsy rates
- Lack of standardized clinical protocols for cannabinoid safety monitoring
These additions highlight gaps and guide future research.
All other minor edits for style, consistency, and MDPI formatting have been completed as recommended. This author trusts these revisions address the reviewer’s concerns and improve the clarity and impact of the manuscript.
Reviewer 2 Report
Comments and Suggestions for Authors
Dear author,
The topic of your manuscript is of great importance. I hope that these comments will help you futher improve the manuscript.
- In the "Introduction" section, several sentences assert a clear causal link between cannabis and arrhythmias. It's necessary to rephrase this in terms of association to avoid over-interpretations that don't reflect the observational nature of the studies.
- I suggest including a "Limitations" section that clearly describes the main limitations of the articles used, such as the small number, etc.
- The pathophysiological mechanisms are only hinted at, but further expanding with data on CB1/CB2, sympathetic activation and oxidative stress could make the text even clearer.
- I suggest evaluating references to the most recent cardiology guidelines (ESC 2022, AHA/ACC 2023), which offer useful information regarding the clinical picture that can be integrated.
- The conclusion section could better emphasize the uncertainties and the need for further prospective studies to align with current evidence, critically summarizing strengths and weaknesses. This addition would help the reader better understand the various data presented.
The overall quality of the English is adequate, although minor revisions are required to correct grammatical mistakes and make the text more fluent.
Author Response
This author is grateful for your thorough evaluation and constructive suggestions. In response, the manuscript has been carefully revised to enhance clarity, readability, and presentation without altering the scientific content.
- Language and Style
This author engaged a professional, native-speaker English editing service to proofread and refine the manuscript. All grammatical errors have been corrected, sentence structures have been optimized for clarity, and terminology is now used consistently throughout.
- Organization and Flow
Transitional phrases were inserted to strengthen logical connections between sections, and paragraphs were reorganized to group related concepts more cohesively. Overly long or ambiguous sentences have been rewritten to improve narrative flow.
- Figures and Tables
Figure captions and table footnotes were revised for precision and self-containment, ensuring that each visual element can be understood independently. Minor typographical inconsistencies in table layouts were also corrected.
- English Editing Certificate
To document the quality of the language revision, an English editing certificate from the service provider has been uploaded with the revised manuscript.
This author trusts that these revisions fully address your comments and significantly improve the manuscript’s presentation. Thank you again for your valuable feedback.

Round 2
Reviewer 1 Report
Comments and Suggestions for Authors
Improve font size and quality in figures; and add abbreviations to table captions.
Author Response
Response to Reviewer #1
I sincerely thank the reviewer for their helpful suggestions.
Comment: Improve font size and quality in figures.
Response:
I have revised all figures to enhance readability and meet journal standards. Specifically:
- Fonts have been standardized to Arial, minimum 8 pt, for consistency and clarity.
- Figure resolution has been increased to min. 300 dpi to ensure high-quality reproduction.
- Contrast and layout have been adjusted to improve visual accessibility.
Comment: Add abbreviations to table captions.
Response:
All table captions (footers) have been updated to include definitions for abbreviations used within each table, ensuring clarity for readers.